# Revisiting the Bethe-Hessian: Improved Community Detection in Sparse Heterogeneous Graphs

**Lorenzo Dall'Amico**
GIPSA-lab, UGA, CNRS, Grenoble INP
lorenzo.dall-amico@gipsa-lab.fr

**Romain Couillet**
GIPSA-lab, UGA, CNRS, Grenoble INP
L2S, CentraleSupélec, University of Paris Saclay

**Nicolas Tremblay**
GIPSA-lab, UGA, CNRS, Grenoble INP

## Abstract

Spectral clustering is one of the most popular, yet still incompletely understood, methods for community detection on graphs. This article studies spectral clustering based on the Bethe-Hessian matrix $H_r = (r^2 - 1)I_n + D - rA$ for sparse heterogeneous graphs (following the degree-corrected stochastic block model) in a two-class setting. For a specific value $r = \zeta$, clustering is shown to be insensitive to the degree heterogeneity. We then study the behavior of the informative eigenvector of $H_\zeta$ and, as a result, predict the clustering accuracy. The article concludes with an overview of the generalization to more than two classes along with extensive simulations on synthetic and real networks corroborating our findings.

## 1 Introduction

Network theory studies the interaction of connected systems of agents. Real networks tend to be structured in affinity classes and the problem of clustering consists in retrieving these unknown classes from the observed network pairwise interactions [1]. Belief propagation (BP) is an efficient way to reconstruct communities and – under certain conditions (see [2]) – was proved to give *optimal* reconstruction. On the negative side, BP suffers from a possibly long convergence time and a non-trivial implementation. Among the alternative clustering algorithms, spectral techniques proved particularly efficient in terms of speed and analytical tractability [3, 4, 5, 6]. In the dense regime, in particular, where the average node degree scales like the size of the network, random matrix theory [4, 7, 8] manages to predict the asymptotic spectral clustering performances and to identify transition points beyond which asymptotic non trivial classification is achievable. This is however not the typical condition for real networks that tend instead to be *sparse*. For a graph $\mathcal{G}(\mathcal{V}, \mathcal{E})$ with $|\mathcal{V}| = n$ nodes, the condition of sparsity means that the average degree $d$ does not depend on the size of the network and in particular $d \ll n$.

Both standard spectral clustering methods and their associated random matrix asymptotics collapse in this regime. As an answer, many intuitions emerged from statistical physics and led to important seminal steps. Notably, two deeply connected matrices recently proved to overcome the problem of sparsity: the $n \times n$ Bethe-Hessian [9] $H_r$ with $r \in \mathbb{R}$ a parameter to be fixed – the study of which is the object of the present article–, and the non symmetric non backtracking operator $B \in \{0,1\}^{2|\mathcal{E}| \times 2|\mathcal{E}|}$ [10]. Both matrices were introduced and studied under the homogeneous degree stochastic block model (SBM). Narrowing to the case of two communities it was proved both experimentally and theoretically [11, 2, 12, 13] that, if there exists an algorithm able to detect communities better then random guess, then these two matrices can be used to give non-trivial node partition. It is said that both algorithms work *down to the detectability threshold*.

However, real networks are rarely homogeneous and typically follow a power law degree distribution [14]. The results of [15, 16] generalize the above studies to heterogeneous networks, generated by degree-corrected stochastic block models (DC-SBM) [17] and suggest that both $B$ and $H_r$ provide also in this case non trivial clustering down to the detectability threshold. Yet, a precise characterization of their behavior and performances is still lacking; the present article shows that some aspects of the behavior of $B$ and $H_r$ have indeed been overlooked.

Spectral clustering in sparse heterogeneous networks has also been tackled using various regularized Laplacian matrices [18, 19, 20] but, to our knowledge, these are not proved to operate down to the detectability threshold. These structurally different methods are discussed in concluding remarks.

The main message of the present communication is that, under a DC-SBM setting, the choice of $r$ in $H_r$ proposed in [9] for the SBM setting is suboptimal. We propose and theoretically support an improved parametrization $r = \zeta$ that allows the Bethe-Hessian $H_\zeta$ to efficiently detect communities in sparse and heterogeneous graphs. In detail, under the DC-SBM setting, a) we propose a spectral algorithm on $H_\zeta$ which performs efficiently down to the detectability threshold, with an informative eigenvector not tainted by the degree distribution (unlike in [9]); b) the algorithm is generalized to $k$-class clustering with a consistent estimation procedure for $k$; c) substantial performance improvements on the originally proposed Bethe-Hessian are testified by simulations on synthetic and real networks.

The remainder of the article is organized as follows: Section 2 argues on the optimal value $r = \zeta$ for $H_r$ and, based on heuristic arguments, studies the behavior of the informative eigenvector of $H_\zeta$, concluding with an explicit expression of the clustering performance; Section 3 provides an unsupervised method to estimate $\zeta$, drawing on connections with the non-backtracking matrix $B$; Section 4 extends the algorithm to a $k$-class scenario; numerical supports are then provided in Section 5 on both synthetic and real networks; concluding remarks close the article.

**Reproducibility.** A Python implementation of the proposed algorithm along with codes to reproduce the results of the article are available at lorenzodallamico.github.io/codes.

## 2 Model and Main Results

### 2.1 Model setting

Consider an undirected binary graph $\mathcal{G}(\mathcal{E}, \mathcal{V})$, with nodes $\mathcal{V} = \{1, \dots, n\}$ ($|\mathcal{V}| = n$) and edges $\mathcal{E} \subset \mathcal{V} \times \mathcal{V}$ ($|\mathcal{E}| = m$). Let $\boldsymbol{\sigma} \in \{-1, 1\}^n$ be the vector of class labels, both classes being of equal size (i.e., $\sum_i \sigma_i = 0$), and $C = \left( \begin{smallmatrix} c_{\text{in}} & c_{\text{out}} \\ c_{\text{out}} & c_{\text{in}} \end{smallmatrix} \right)$. These assumptions are meant to set the problem in a more readable symmetric scenario. Section 4 extends the results to multiple classes of possibly different sizes. In order to account both for sparsity and heterogeneity, we consider the DC-SBM as a generative model for $\mathcal{G}$. Denoting $A \in \{0, 1\}^{n \times n}$ the adjacency matrix defined by $A_{ij} = 1_{(i,j) \in \mathcal{E}}$, the DC-SBM generates edges independently according to:

$$\mathbb{P}(A_{ij} = 1 | \sigma_i, \sigma_j, \theta_i, \theta_j) = \theta_i \theta_j \frac{C_{\sigma_i, \sigma_j}}{n}, \tag{1}$$

where $\boldsymbol{\theta} = (\theta_1, \dots, \theta_n)$ is the vector of random intrinsic connection "probabilities" of each node. The $\theta_i$'s are assumed i.i.d. and independent of the class labels, and we impose $\mathbb{E}[\theta_i] = 1$, $\mathbb{E}[\theta_i^2] = \Phi$. The $1/n$ term bounds the degree of each node to an $n$-independent value, making the network sparse. Denoting $c = (c_{\text{in}} + c_{\text{out}})/2$, the detectability condition [16] reads:

$$\alpha \equiv \frac{c_{\text{in}} - c_{\text{out}}}{\sqrt{c}} \geq \frac{2}{\sqrt{\Phi}} \equiv \alpha_c. \tag{2}$$

For $\alpha < \alpha_c$, no algorithm can partition the nodes better than by random guess. Letting $D = \text{diag}(A\mathbb{1})$ be the degree matrix, the Bethe-Hessian is defined as

$$H_r = (r^2 - 1)I_n + D - rA, \quad r \in \mathbb{R}. \tag{3}$$

This matrix was originally proposed in [9] for $r = \sqrt{c\Phi}$, which asymptotically provides non trivial clustering down to the *detectability threshold* (for $\alpha > \alpha_c$). The informative eigenvector of $H_r$ is associated with the second smallest eigenvalue and we denote it $\boldsymbol{x}_r^{(2)}$. The components of $\boldsymbol{x}_{\sqrt{c\Phi}}^{(2)}$ are however strongly tainted by the $\theta_i$'s, sensibly altering the algorithm performance.

We show here that for $\alpha \geq \alpha_c$ there exists a value $\zeta \leq \sqrt{c\Phi}$ for which the components of the second eigenvector $\boldsymbol{x}_\zeta^{(2)}$ of $H_\zeta$ align to the labels irrespective of the $\theta_i$'s, thus largely improving the algorithm performance while maintaining detectability down to the threshold.

## 2.2 Informative eigenvector of $H_r$

In the sequel we assume that: (i) being sparse, we can locally approximate the graph by a tree [21] and therefore $\mathbb{P}(\boldsymbol{\sigma}_{\partial_i}|\sigma_i) \simeq \prod_{j \in \partial_i} \mathbb{P}(\sigma_j|\sigma_i)$, with $\partial_i$ the neighbourhood of $i$; (ii) $n \to \infty$ and $c$ is bounded by an $n$-independent value while being arbitrarily larger than one, i.e., $n \gg c \gg 1$.

For ease of notation we work here with $D - rA$ rather than $H_r$, both having the same eigenvectors. The core of our proposed method lies in the following observation, related to the action of $H_r$ on $\boldsymbol{\sigma}$:

$$[(D - rA)\boldsymbol{\sigma}]_i = d_i\sigma_i\left[1 - r\left(\frac{|\partial_i^{(s)}|}{d_i} - \frac{|\partial_i^{(o)}|}{d_i}\right)\right] \tag{4}$$

where $|\partial_i^{(s)}|$ (resp., $|\partial_i^{(o)}|$) stands for the number of neighbors of $i$ belonging to the same (resp., opposite) class as $i$. We show next that a proper choice of $r$ can annihilate the right-hand side of (4) "on average" or whenever the typical degrees $d_i$ are not too small, turning (4) into an eigenvector equation. To this end, we need to quantify the random variables $|\partial_i^{(s)}|$ and $|\partial_i^{(o)}|$.

From a Bayesian perspective, $\boldsymbol{\sigma}$ and $\boldsymbol{\theta}$ are unknown parameters and $A$ (and thus $d_i$) known realizations. We may thus write

$$\mathbb{P}(\sigma_i|\sigma_j, A_{ij} = 1) = \frac{\mathbb{P}(\sigma_i, \sigma_j|A_{ij} = 1)}{\mathbb{P}(\sigma_j|A_{ij} = 1)} = 2\iint \mathbb{P}(\sigma_i, \sigma_j, \theta_i, \theta_j|A_{ij} = 1)d\theta_i d\theta_j$$

$$\propto \iint \mathbb{P}(A_{ij} = 1|\sigma_i, \sigma_j, \theta_i, \theta_j)\mathbb{P}(\sigma_i, \sigma_j, \theta_i, \theta_j)d\theta_i d\theta_j \propto C(\sigma_i, \sigma_j),$$

where we used the facts that the classes are of equal size ($\mathbb{P}(\sigma_i)$ is constant), and the $\theta_i$ are i.i.d., independent of the classes with $\mathbb{E}[\theta_i] = 1$. Normalizing, one finally obtains $\mathbb{P}(\sigma_i|\sigma_j, A_{ij} = 1) = C(\sigma_i, \sigma_j)/(c_{\text{in}} + c_{\text{out}})$, which is independent of the degree distribution. We further know that $|\partial_i^{(s)}| + |\partial_i^{(o)}| = d_i$, which is a deterministic observation. Given the locally tree-like structure of the graph, neighbors of the same node are conditionally independent – see (i) – so that $|\partial_i^{(s)}|$ is the sum of $d_i$ i.i.d. Bernoulli random variables with parameter $p = c_{\text{in}}/(c_{\text{in}} + c_{\text{out}})$. We thus obtain

$$\mathbb{E}[[(D - rA)\boldsymbol{\sigma}]_i \mid A] = d_i\sigma_i\left(1 - r\frac{c_{\text{in}} - c_{\text{out}}}{c_{\text{in}} + c_{\text{out}}}\right). \tag{5}$$

This equation suggests that, for the expectation of (4) to be an eigenvector equation in the large (but finite) $d_i$ regime, $r$ should be taken equal to

$$r = \frac{c_{\text{in}} + c_{\text{out}}}{c_{\text{in}} - c_{\text{out}}} = \frac{2\sqrt{c}}{\alpha} \equiv \zeta_\alpha. \tag{6}$$

with $\alpha$ as in (2) the proper control parameter for the clustering problem (as shown e.g., in [7, 15, 16, 22]). For simplicity of notation the dependence on $\alpha$ of $\zeta = \zeta_\alpha$ will be made explicit only when relevant. Intuitively, this calculus suggests that $\zeta$ is the only value of $r$ that ensures that $H_r$ has an informative eigenvector not significantly tainted by the degree distribution. This claim is supported by the following two remarks.

**Remark 1** (Consistency of $\zeta$ for trivial classification). *In the limit of trivial clustering where $c_{\text{out}} \to 0$, $|\partial_i^{(s)}|$ and $|\partial_i^{(o)}|$ tend to their mean. In particular, for $c_{\text{out}} = 0$, $\zeta = 1$ and $(D - \zeta A)\boldsymbol{\sigma} = (D - A)\boldsymbol{\sigma} = 0$, so that $\boldsymbol{\sigma}$ is an exact eigenvector of $H_{\zeta=1}$ associated with its zero eigenvalue.*

**Remark 2** (Mapping to Ising). *The original intuition behind the Bethe-Hessian matrix arises from a mapping of the community labels into the spins of a Ising Hamiltonian. The "temperature-related" parameter $r$ guarantees a correct mapping only for $r = \zeta$. This is elaborated in details in Section A of the supplementary material.*

Although one commonly assumes an assortative model for the communities, by which $c_{\text{in}} > c_{\text{out}}$, the Bethe-Hessian matrix is oblivious of the sign of $c_{\text{in}} - c_{\text{out}}$.

**Remark 3** (Disassortative networks). *The case where $c_{\text{out}} > c_{\text{in}}$ does not invalidate the above analysis which results in $\zeta < 0$. Clustering with $H_\zeta$ is thus also valid in disassortative networks.*

In practice, for a given (non averaged) realization of the $\sigma_i$'s, $\boldsymbol{\sigma}$ is not an exact eigenvector of $H_\zeta$. By a perturbation analysis around $\boldsymbol{\sigma}$, we next analyze the behavior of the corresponding informative eigenvector of $H_\zeta$ and theoretically predict the overlap performance.

## 2.3 Performance Analysis

To generalize the averaged analysis of (5), we perturb $\boldsymbol{\sigma}$ by a "noise" term $\boldsymbol{\delta}$ and write $\boldsymbol{x}_\zeta^{(2)} \equiv \boldsymbol{\sigma} + \boldsymbol{\delta}$. Since $\zeta$ is however maintained, the associated eigenvalue of $D - \zeta A$, which is zero in (5), now possibly deviates from zero; this eigenvalue is denoted $\lambda_\alpha$, i.e.,

$$(D - \zeta_\alpha A)(\boldsymbol{\sigma} + \boldsymbol{\delta}) = \lambda_\alpha(\boldsymbol{\sigma} + \boldsymbol{\delta}). \tag{7}$$

From Remark 1, we already know that $\lim_{\alpha \to \sqrt{2c_{\text{in}}}} \lambda_\alpha = 0$.

In the following, expectations are taken for a fixed realization of the network, *i.e.* $\mathbb{E}[\cdot] \equiv \mathbb{E}[\cdot|A]$. Writing $|\partial_i^s| = \mathbb{E}[|\partial_i^s|] + \Delta_i$ and $|\partial_i^o| = \mathbb{E}[|\partial_i^o|] - \Delta_i$, where we exploited the relation $|\partial_i^s| + |\partial_i^o| = d_i$, we obtain:

$$[(D - \zeta_\alpha A)(\boldsymbol{\sigma} + \boldsymbol{\delta})]_i = -2\zeta_\alpha \sigma_i \Delta_i + d_i \delta_i - \zeta_\alpha \sum_{j \in \partial_i} \delta_j. \tag{8}$$

The random variable $\Delta_i$ is a sum of $d_i$ independent (centered) Bernoulli random variables, tending in the large $c$ limit to a zero mean Gaussian, i.e.,

$$\Delta_i \sim \mathcal{N}(0, d_i c_{\text{in}} c_{\text{out}}/(c_{\text{in}} + c_{\text{out}})^2) \equiv \mathcal{N}(0, d_i f_\alpha^2/\zeta_\alpha^2), \quad f_\alpha \equiv \frac{\sqrt{c_{\text{in}} c_{\text{out}}}}{c_{\text{in}} - c_{\text{out}}} = \frac{1}{\alpha}\sqrt{c - \frac{\alpha^2}{4}}. \tag{9}$$

Our analysis of (8) relies on the following claim that we shall justify next.

**Assumption 1.** *The random variables $\delta_i$, $1 \le i \le n$, are distributed as $\delta_i \sim \mathcal{N}(-\mu_\alpha \sigma_i, f_\alpha^2 \beta_i^2)$ for some $\mu_\alpha \in \mathbb{R}$ depending on $\alpha$ only, and $\beta_i \in \mathbb{R}$ depending on $i$ only. Besides, the $\delta_i$'s are "weakly dependent" in the sense that $\mathbb{E}[\delta_i \delta_j] = \mathbb{E}[\delta_i]\mathbb{E}[\delta_j] + O(1/c)$.*

The elements of Assumption 1 rely on the following observations:

- *Weak dependence*: This claim follows from the weak dependence of the $\Delta_i$'s, which results from the sparse (and thus locally tree-like) nature of the graph.
- *Gaussianity*: The right-hand side of (8) features 3 random variables, the leftmost being Gaussian and rightmost the sum of $d_i$ variables tending to an (asymptotically independent) Gaussian. It is thus reasonable that $\delta_i$ be Gaussian (so to ensure (7)) yet not independent of $\Delta_i$ or $\sum_{j \in \partial_i} \delta_j$.
- *Mean of $\delta_i$*: The symmetry of the problem at hand (equal class sizes, same affinity $c_{\text{in}}$ for each class), along with the fact that the right-hand side of (4) vanishes in its first order approximation in $d_i$, suggest that the mean of $\delta_i$ does not depend in the first order on $d_i$ but only on $\sigma_i$. The amplitude of the mean then depends on $\alpha$ characterized here through $\mu_\alpha$.
- *Variance of $\delta_i$*: The variance appears as the product of two terms: one that depends on $i$ ($\beta_i$) and one that depends on $\alpha$. This follows from assuming that the fluctuations of $\delta_i$ follow the fluctuations of $\Delta_i$ for which the variance is similarly factorized.

Imposing the norm of the eigenvector $\boldsymbol{x}_\zeta^{(2)} = \boldsymbol{\sigma} + \boldsymbol{\delta}$ to be constant with respect to $\alpha$ and the boundary condition $\mu_{\alpha_c} = 1$ (i.e., there is no information about the classes at the detectability threshold), we find the following explicit expressions for $\mu_\alpha$ and $\beta_i$.

$$1 - \mu_\alpha = \sqrt{\frac{c\Phi - \zeta_\alpha^2}{c\Phi - 1}}, \quad \beta_i = \frac{2}{\sqrt{d_i}}.$$

Details are provided in Section B of the supplementary material. Figure 1-(a) supports the analysis by comparing this prediction to simulations for a synthetic network with power law degree distribution.

The previous line of argument provides a large dimensional approximation for the performance of spectral clustering based on the eigenvector $\boldsymbol{x}_\zeta^{(2)}$. The performance measure of interest is the

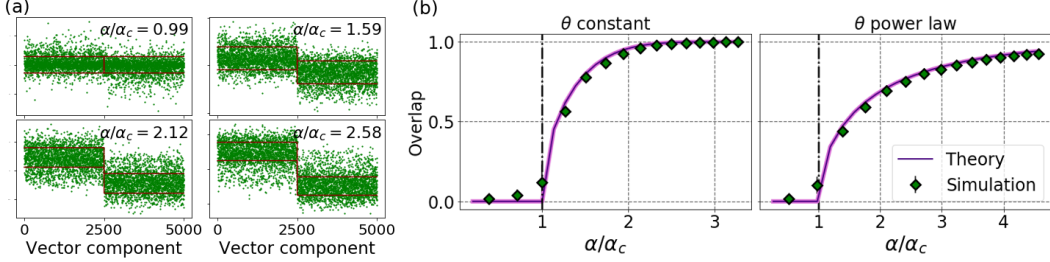

Figure 1: (a) Theoretical values of mean and variance (red line indicates $1 - \mu_\alpha \pm 2f_\alpha/\sqrt{c}$) vs simulation (green dots) for power-law distributed $\theta_i$'s ($\theta_i \sim Z^{-1}[\mathcal{U}(3, 10)]^4$). (b) Theoretical (10) vs simulated overlap, averaged over 10 realizations, for $\theta_i$ constant (left), and power-law distributed (right). For both figures, $n = 5000$, $c_{\text{out}} = 6$, $c_{\text{in}} = 7 \to 36$.

*overlap*, defined as $\text{Ov} \equiv 2 \max_{\mathcal{P}_{\hat{\sigma}}} \left[ \frac{1}{n} \sum_{i=1}^n \delta_{\sigma_i, \hat{\sigma}_i} - \frac{1}{2} \right]$ where $\hat{\sigma}$ denotes the vector of estimated labels, $\mathcal{P}_{\hat{\sigma}}$ the set of permutations of the labels, and $\delta$ the Kronecker symbol ($\delta_{ij} = 1$ if $i = j$, and 0 otherwise). In this particularly symmetric setting *only* $\hat{\sigma}_i = \text{sign}[(\boldsymbol{x}_\zeta^{(2)})_i]$ where $\text{sign}$ is the sign function. (Remark 5 underlines the necessity not to cluster based on sign in asymmetric scenarios). From the expression of $\mu_\alpha$ and $\beta_i$, we find that, conditionally to $A$,

$$\mathbb{E}[\text{Ov}] \simeq \frac{1}{n} \sum_{i=1}^n \text{erf} \left[ \sqrt{\frac{\alpha^2 d_i}{8c - 2\alpha^2} \left( \frac{c\Phi - \zeta_\alpha^2}{c\Phi - 1} \right)} \right] \tag{10}$$

(proof details are provided in Section B of the supplementary material). Figure 1-(b) compares the prediction of Equation (10) to simulations on networks with $\theta_i = 1$ constant (left) or power-law distributed (right). The observed match on this $5\,000$-node synthetic network is close to perfect.

As a side remark, our analysis reveals an interesting connection between $H_\zeta$ and $D^{-1}A$.

**Remark 4** (Relation to the random walk Laplacian). *Similar to $A$, $D - A$, and $D^{-\frac{1}{2}}AD^{-\frac{1}{2}}$, the matrix $D^{-1}A$ is claimed inappropriate as a spectral community detection matrix for sparse graphs. This is in fact a slight overstatement: as already observed in [20], as the graph under study gets sparser, $D^{-1}A$ still possesses one or possibly more informative eigenvectors, however not necessarily corresponding to dominant isolated eigenvalues (it was in particular noted that for the real network polblogs [23] the informative eigenvector is associated to the third and not the second largest eigenvalue). This observation is easily explained in our analysis framework. Similar to our derivation for $D - \zeta A$, the average action of $D^{-1}A$ on the class vector $\boldsymbol{\sigma}$ reads $\mathbb{E}[[D^{-1}A\boldsymbol{\sigma}]_i|A] = \sigma_i/\zeta$ and thus, for large $d_i$, $\boldsymbol{\sigma}$ is a close eigenvector to $D^{-1}A$, correctly predicting the existence of an informative eigenvalue also for this matrix. However, the associated eigenvalue $1/\zeta$ decays with increasing $\zeta$ and thus with harder detection tasks, hence explaining why the informative eigenvectors are associated with eigenvalues found deeper into the spectrum of $D^{-1}A$.*

## 3 Estimating $\zeta$

While $r = \zeta$ is more appropriate a choice than $r = \sqrt{c\Phi}$, $\zeta$ is not readily accessible (as it depends on $c_{\text{in}} - c_{\text{out}}$), unlike $\sqrt{c\Phi}$ that is easily estimated from the $d_i$'s. To estimate $\zeta$, we elaborate on the deep relations between the Bethe Hessian $H_r$ and the non-backtracking operator $B \in \mathbb{R}^{2|\mathcal{E}| \times 2|\mathcal{E}|}$ defined, for all $(ij), (lm) \in \mathcal{E}_D$ the set of directed edges of $\mathcal{G}$, as $B_{(ij)(lm)} = \delta_{jl}(1 - \delta_{im})$.

When $r$ is an eigenvalue of $B$, then $\det H_r = 0$ [11, 24]. This is convenient as $B$ only has a few isolated real eigenvalues ($B$ is non symmetric) that can send the associated isolated eigenvalues of $H_r$ to zero. This provides us with two alternative methods to estimate $\zeta$.

### 3.1 Exploiting the eigenvalues outside the bulk of $B$

It is proved in [15] that, for the DC-SBM and beyond the phase transition ($\alpha > \alpha_c$), the eigenvalues $\gamma_1, \ldots, \gamma_{2m}$ of $B$, decreasingly sorted in modulus, satisfy in the large $n$ setting: $\gamma_1 \to \Phi(c_{\text{in}} + c_{\text{out}})/2$, $\gamma_2 \to \Phi(c_{\text{in}} - c_{\text{out}})/2 > \sqrt{\gamma_1}$ and, for $i > 2$, $\limsup_n |\gamma_i| \leq \sqrt{\gamma_1}$, almost surely.

Since $\zeta = \lim_n \gamma_1/\gamma_2$, denoting $\nu_i(r)$ the eigenvalues of $H_r$ sorted in *increasing* order, this result conveys the following first method to estimate $\zeta$.

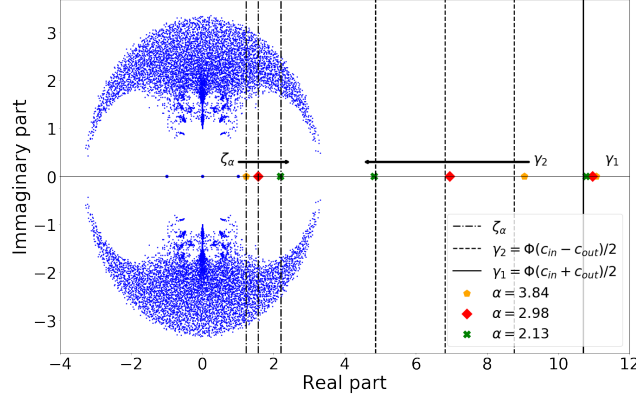

Figure 2: Superposed spectra of $B$ for 3 values of $\alpha$: $n = 4000$, $c_{\text{in}} = 12, 11, 10$ and $c_{\text{out}} = 1, 2, 3$ ($c_{\text{in}} + c_{\text{out}}$ is fixed); $\theta$ with power law distribution; all eigenvalues displayed in blue except top three dominant real displayed in colors for each $(c_{\text{in}}, c_{\text{out}})$ pair.

**Method 1** (First estimation of $\zeta$). *Under the previous notations $\zeta \simeq \gamma_1/\gamma_2$. The eigenvalues $\gamma_1$ and $\gamma_2$ of $B$ can be estimated by a line search over $r \in (\sqrt{\rho(B)}, \infty)$ on changing signs of $\nu_1(r)$ and $\nu_2(r)$ that correspond to $r = \gamma_1$ and $r = \gamma_2$, respectively.*[1]

### 3.2 Exploiting the eigenvalues inside the bulk of $B$

The matrix $B$ can be obtained from the linearization of the belief propagation (BP) equations (see [10] for details). In particular, the linear expansion to first order of the beliefs around their fixed points yields $Bw \simeq \zeta w$. According to this argument, one expects the matrix $B$ to have a real eigenvalue equal to $\zeta$ with[2] $\zeta \leq \sqrt{c\Phi}$. Figure 2 visually emphasizes this eigenvalue for three different values of $\alpha$, maintaining $c$ constant. The matrix $B$ thus has four eigenvalues inside its main bulk: $-1, 0, 1$ and $\zeta$. As the community detection problem gets harder, both $\zeta$ and $\gamma_2$ shift towards the edge of the bulk (from the left for the former and from the right for the latter) and then meet exactly at $\sqrt{c\Phi}$ when $\alpha = \alpha_c$. Then, for $\alpha < \alpha_c$, they reach the complex part of the bulk.

More fundamentally, simulations further suggest that the eigenvector associated with the null eigenvalue of $H_\zeta$ is precisely $\boldsymbol{x}_\zeta^{(2)} = \boldsymbol{\sigma} + \boldsymbol{\delta}$ studied in Section 2.3. This indicates that the informative eigenvalue $\lambda_\alpha$ of $D - \zeta_\alpha A = H_{\zeta_\alpha} - (\zeta_\alpha^2 - 1)I_n$ in Equation (7) coincides with $-(\zeta_\alpha^2 - 1)$. It further explains why $H_{\sqrt{c\Phi}}$, initially proposed in [9], works well close to the detectability threshold as $\zeta \to \sqrt{c\Phi}$ when $\alpha \to \alpha_c$. We thus expect most of the improvement of the choice $r = \zeta$ to emerge in the easier scenarios.

Note that, as was already observed in [9], if $|r| > 1$, then the eigenvalues of the bulk of $H_r$ are strictly positive for $|r| \neq \sqrt{c\Phi}$. As a consequence, $\boldsymbol{x}_\zeta^{(2)}$ is necessarily isolated when $\alpha > \alpha_c$ and so spectral clustering on $H_\zeta$ works down to the detectability threshold. To the best of our knowledge, this property is not formally proved, but we point out that it agrees with the shape of the spectrum of $B$: if the bulk of $H_r$ was negative for some $|r| > 1$, then there would be a 'continuum' of real eigenvalues in $[1, \sqrt{c\Phi}]$ if $r > 1$ (in the assortative case). As this is not the case, the smallest eigenvalue in the bulk of $H_r$ cannot be negative.

**Claim 1** (Informative eigenvalue of $H_{\zeta_\alpha}$). *The eigenvalue associated to the informative eigenvector of $H_{\zeta_\alpha}$ is equal to zero. Equivalently, the eigenvalue $\lambda_\alpha$ associated to the informative eigenvector of $D - \zeta_\alpha A$ is given by $\lambda_\alpha = -(\zeta_\alpha^2 - 1) = -4f_\alpha^2$ which vanishes for $c_{\text{out}} \to 0$.*

This claim gives rise to a second method to estimate $\zeta$.

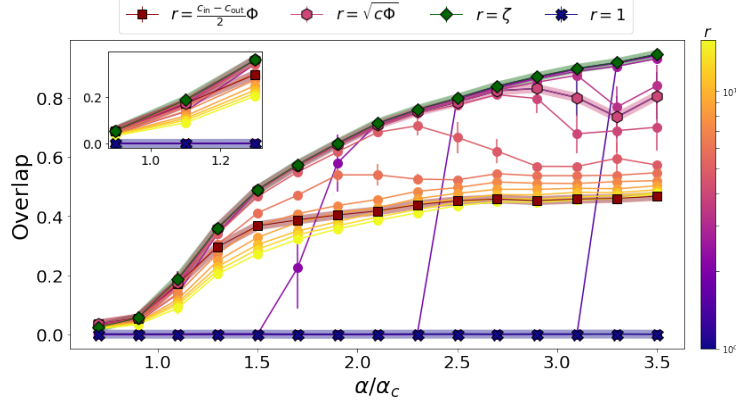

Figure 3: Overlap comparison as a function of $\alpha$, using the second smallest eigenvector of $H_r$, for different values of $r$. In color code the values of $r$ ranging from $r = 1$ (blue) to $r = c\Phi$ (yellow). The red squares indicate $r = (c_{\text{in}} - c_{\text{out}})\Phi/2$, that is equivalent to clustering with the matrix $B$ [10], the purple hexagons represent the Bethe-Hessian of [9], the green diamonds are the proposed Algorithm 1 and the blue crosses are the graph Laplacian. In the top left corner, a zoom of the overlap close to the transition. For these simulations, $n = 5000$, $c_{\text{in}} : 15 \to 9.4$, $c_{\text{out}} : 1 \to 6.6$ (while keeping $c$ fixed), $\theta_i \sim [\mathcal{U}(3, 10)]^4$.

**Method 2** (Second estimation of $\zeta$). *Under the previous notations $\nu_2(\zeta) = 0$. The parameter $\zeta$ then corresponds to the position of change of sign of $\nu_2(r)$ in the set $r \in (1, \sqrt{\rho(B)})$.*

## 4 Extension to multiple uneven-sized classes

The analysis performed in the previous sections is resilient to heterogeneous degree distributions and can be generalized to $k$ uneven-sized classes, with last clustering step by *k-means*. To this end, let $\Pi \in \mathbb{R}^{k \times k}$ be diagonal with $\Pi_{ii}$ the fraction of nodes in class $i$ and assume $C\Pi \mathbb{1} = c\mathbb{1}$. This assumption is a standard hypothesis [10, 22, 11, 25] which ensures that the averaged node connectivity is independent of the class. For $1 \leq p \leq k$, let $(\tau_p, \boldsymbol{v}^{(p)})$ be the $p$-th largest eigenpair of $C\Pi$, and $\boldsymbol{u}^{(p)} \in \mathbb{R}^n$ defined as $u_i^{(p)} = v_{\ell_i}^{(p)} \; \forall \; 1 \leq i \leq n$ for $\ell_i$ the class of node $i$. The vector $\boldsymbol{u}^{(p)}$ contains plateaus with heights corresponding to the values of $\boldsymbol{v}^{(p)}$. Repeating the arguments of Section 2 (see details in Section C of the supplementary material), we obtain $k$ choices for $r$:

$$\mathbb{E}[[(D - rA)\boldsymbol{u}^{(p)}]_i] = d_i u_i^{(p)} \left[1 - r\frac{\tau_p}{c}\right] \qquad \text{and thus} \quad r = \frac{c}{\tau_p} \equiv \zeta_p, \quad 1 \leq p \leq k. \tag{11}$$

Since the largest eigenpair $(c, \mathbb{1})$ of $C\Pi$ is not informative of the class structure, only the $k - 1$ next largest eigenvectors $\boldsymbol{v}^{(p)}$ of $C\Pi$ are informative. The vector $\boldsymbol{u}^{(p)}$ (corresponding to the $p$-th largest eigenvalue $\tau_p$) is in one-to-one mapping with $\boldsymbol{v}^{(p)}$ and corresponds to the $p$-th smallest value of $\zeta_p = c/\tau_p$. Considering $r = \sqrt{c\Phi}$, all the informative eigenvalues of $H_r$ are negative [9]. By decreasing $r$ they progressively become positive: for $r = \zeta_k$ (the largest among $\zeta_p$) the $k$-th smallest eigenvalue is the first to hit zero. By further decreasing $r$, all the informative eigenvalues follow, until $r = \zeta_1 = 1$ for which the smallest eigenvalue is null. We conclude that $\boldsymbol{u}^{(p)}$ is associated with the $p$-th smallest eigenvector $\boldsymbol{x}_{\zeta_p}^{(p)}$ of $H_{\zeta_p}$.

Method 1 and Method 2 both generalize to this scenario. In particular the outer eigenvalues of $B$ converge as $\gamma_p \to \tau_p\Phi$ and the linearization of BP retrieves the fixed points as $\zeta_p = c/\tau_p$.

For $k > 2$, the value $r = \sqrt{c\Phi}$ still plays an important role. It was chosen in [9] because, asymptotically, for this value of $r$ *only* the informative eigenvalues of $H_{\sqrt{c\Phi}}$ are negative. The number of classes is then directly obtained from counting the number of negative eigenvalues of $H_{\sqrt{c\Phi}}$. The relation between $H_r$ and $B$ further guarantees that the number of isolated eigenvalues of $B$ (hence of $H_r$) is asymptotically equal to the number of detectable classes.

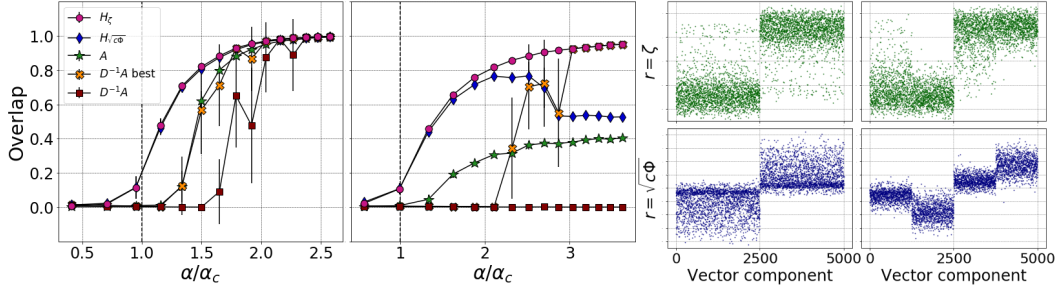

Figure 4: (a) Comparison of spectral clustering for $\theta_i = 1$ (left) and with power law distribution $\theta_i \sim Z^{-1}[\mathcal{U}(3,10)]^4$. "$D^{-1}A$ best" indicates spectral clustering on the best (among the first 25) eigenvector of $D^{-1}A$. Here, $n = 5000$, $c_{\text{out}} = 1$, $c_{\text{in}} = 2 \to 16$. Averaged over 10 samples. The error bars indicate one standard deviation. (b) $\boldsymbol{x}_\zeta^{(2)}$ (top) and $\boldsymbol{x}_{\sqrt{c\Phi}}^{(2)}$ (bottom) for power law distributed $\theta_i$ (left) and for $\theta_i = \theta_0$, $i \leq n/4$ and $n/2 \leq i \leq 3n/4$, and $\theta_i = 4\theta_0$ otherwise (right).

**Remark 5** (On k-means versus signed-based clustering). *Under a symmetric 2-class of even size setting, the classification of the entries of the informative eigenvector of $H_r$ can be performed based on their signs. This sign-based method first does not generalize to more than two or uneven sized classes, where k-means or expectation-maximization based clustering is required. But it also hinders the fact that the eigenvector entries may be quite concentrated around zero (close to $0^+$ or $0^-$ according to the class) and thus* not clustered, *a situation where k-means has no discriminative power.*

*Simulations (and reported results in [9] based on signs rather than k-means) suggest that the informative eigenvector of $H_{\sqrt{c\Phi}}$ precisely suffers this condition. We have demonstrated here instead that the informative eigenvector of $H_\zeta$ has the convenient feature of being genuinely clustered.*

## 5  Experimental validation

Our results can be summarized by Algorithm 1, where we recall that $\nu_p(r)$ is the $p$-th smallest eigenvalue of $H_r$ and where $\boldsymbol{x}_r^{(p)}$ indicates the corresponding eigenvector.

Figure 3 depicts the overlap, as a function of $\alpha$, of the output of a two-class $k$-means on the informative eigenvector of $H_r$, for different values of $r$, ranging from 1 to $c\Phi$. When $\alpha$ is large enough, small values of $r$ lead to better partitions than large values of $r$ that are more affected by degree heterogeneity. However, for $r$ small, the informative eigenvector is not necessarily corresponding to the second smallest eigenvalue, leading to a meaningless partition. On the contrary, larger values of $r$ show isolated eigenvectors also in the "hard regime". We recall that $r = \zeta$ is an $\alpha$-dependent parameter: for $\alpha \to \alpha_c$, $\zeta$ is "large enough" so that the informative eigenvalue is isolated, while for $\alpha \to \sqrt{2c_{\text{in}}}$ it is "small enough" to give good partitions. Also the value of $r = (c_{\text{in}} - c_{\text{out}})\Phi/2$ is $\alpha$-dependent and it corresponds to clustering with $B$ as indicated in [10]. While it gives good partitions very close to the transition, this choice of $r$ seems largely sub-optimal for easier tasks.

Figure 4-(a) compares the overlaps obtained with Algorithm 1 versus related spectral clustering methods based on $H_{\sqrt{c\Phi}}$, $D^{-1}A$ and $A$. Accordingly with Remark 5, *k-means* clustering (rather than sign-based) on the informative eigenvectors is systematically performed. For $\theta_i = 1$, the left display recovers the results of [9], evidencing a strong advantage for $H_r$ versus Laplacian methods. Since the degrees are similar, both $r = \sqrt{c\Phi}$ and $r = \zeta$ induce similar $H_r$ performances. The improvement provided by $H_\zeta$ arises in the right display for power-law distributed $\theta_i$, with most of the gain appearing away from the detection threshold. On both displays is also depicted the performance of $D^{-1}A$ based on its second largest eigenvector and on an oracle choice of the informative eigenvector with maximal overlap. These curves confirm Remark 4 on the non-dominant position of the informative eigenvector of $D^{-1}A$ in hard tasks.[3] Figure 4-(b) depicts the informative eigenvectors of $H_{\sqrt{c\Phi}}$ and $H_\zeta$, demonstrating the negative impact of $\theta_i$ on $H_{\sqrt{c\Phi}}$, in stark contrast with the resilience of $H_\zeta$.

**Algorithm 1** Improved Bethe-Hessian Community Detection

---
1: **Input** : adjacency matrix of undirected graph $\mathcal{G}$
2: Detect the number of classes: $\hat{k} \leftarrow |\{i,\ \nu_i(\sqrt{c\Phi}) < 0\}|$.
3: **for** $2 \leq p \leq \hat{k}$ **do**
4: $\quad \zeta_p \leftarrow r$ such that $\nu_p(r) = 0$
5: $\quad X_p \leftarrow \boldsymbol{x}_{\zeta_p}^{(p)}$
6: Estimate community labels $\hat{\boldsymbol{\ell}}$ as output of $\hat{k}$-class *k-means* on the rows of $X = [X_2, \dots, X_{\hat{k}}]$.
   **return** Estimated number $\hat{k}$ of communities and label vector $\hat{\boldsymbol{\ell}}$.

---

Table 1 next provides a comparison of the algorithm performances on real networks, both labelled and unlabelled, confirming the overall superiority of Algorithm 1, quite unlike $H_{\sqrt{c\Phi}}$ which fails on several examples.[4]

| L | $n$ | $k$ | Alg.1 | $H_{\sqrt{c\Phi}}$ | $A$ | U | $n$ | $k$ | Alg.1 | $H_{\sqrt{c\Phi}}$ | $A$ |
|---|---|---|---|---|---|---|---|---|---|---|---|
| Karate [28] | 34 | 2 | **1.00** | **1.00** | **1.00** | Mail | 1133 | 21 | **0.50** | 0.40 | 0.32 |
| Dolphins [29] | 62 | 2 | **0.97** | 0.87 | 0.65 | Facebook | 4039 | 65 | **0.77** | 0.48 | 0.38 |
| Polbooks [30] | 105 | 3 | **0.77** | 0.74 | 0.57 | Power grid | 4941 | 53 | **0.92** | 0.61 | 0.31 |
| Football [31] | 115 | 12 | **0.92** | **0.92** | **0.92** | Nutella | 6301 | 5 | **0.34** | 0.15 | 0.14 |
| Polblogs [23] | 1221 | 2 | **0.91** | 0.32 | 0.26 | Wikipedia | 7115 | 21 | **0.21** | 0.18 | 0.15 |

Table 1: Performance comparison on real networks. Labelled datasets with $k$ known and overlap comparison: (left). Unlabelled networks [32] with $k$ estimated and modularity comparison. Only assortative features are kept into account.

## 6  Concluding Remarks

Beyond the demonstration of superiority of $H_\zeta$ to $H_{\sqrt{c\Phi}}$, originally proposed in [9], the article provides a consistent understanding of the natural limitations and strengths of the wide class of spectral clustering methods involving combinations of $A$ and $D$.

Yet, other methods, the performances of which cannot always be compared on even grounds, have been proposed in the literature that marginally relate to the present study. This is notably the case of [18] which performs spectral clustering on $L_\tau = (D + \tau I_n)^{-\frac{1}{2}} A (D + \tau I_n)^{-\frac{1}{2}}$ (with a proposed choice $\tau = c$) which aims at neutralizing the deleterious effects of small $d_i$. Although evidently affecting the spectrum (and thus the informative structure) of $A$ by the non-linear normalization, simulations on $L_\tau$ suggest competitive performances to $H_\zeta$ in almost all studied examples. A systematic analysis of this and similarly proposed methods in the literature is clearly called for.

Despite its demonstrated significant performance improvement, Algorithm 1 suffers from a slightly larger computational cost than most competing methods ($O(nk^3)$ instead of the usual $O(nk^2)$ complexity in the case of sparse graph) due to the successive estimations of $\zeta$. We are currently working on improving this computation time.

From a theoretical standpoint, the request for $c \gg 1$ is still inappropriate to many practical networks. A first consequence of smaller values for $c$ is the loss of Gaussianity of the eigenvector entries as already evidenced in Figures 1 and 4 where Gaussianity is clearly lost in the easiest tasks in profit of a "one-sided" distribution. This suggests further improvement of our analysis framework along with the development of algorithms more appropriate than k-means to handle the last clustering step.

### Acknowledgments

This work is supported by the ANR Project RMT4GRAPH (ANR-14-CE28-0006), the IDEX GSTATS Chair at University Grenoble Alpes and by CNRS PEPS I3A (Project RW4SPEC). The authors thank Jean-Louis Barrat for fruitful discussions.

## Footnotes

[1]The spectral radius of the matrix $B$, $\rho(B)$, can be estimated as $\rho(B) \simeq \sum_i d_i^2 / \sum_i d_i$.

[2]This eigenvalue is visible in [10, 11] but not commented.

[3]The low performance of $D^{-1}A$, even in an oracle setting, can be attributed to the high density of eigenvalues in the bulk of the spectrum which induces a "dispersion" of the informative eigenvectors to the eigenvectors associated to neighboring eigenvalues. The class information is thus "spread" across several eigenvectors.

[4]In Table 1, the modularity is defined as $\mathcal{M} = \frac{1}{2|\mathcal{E}|} \sum_{i,j=1}^{n} \left( A_{ij} - \frac{d_i d_j}{2|\mathcal{E}|} \right) \delta(\hat{\ell}_i, \hat{\ell}_j)$, see e.g., [26, 27].

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
