[Supplementary Material]

## Supplementary material

### A Mapping to Ising

As introduced in [1] for the SBM ($\boldsymbol{\theta} = \mathbb{1}_n$), the probability to realize a graph under the sparse DC-SBM hypothesis reads:

$$
\mathbb{P}(A|\boldsymbol{\sigma}, \boldsymbol{\theta}) = \prod_{i,j<i} \left( \theta_i \theta_j \frac{C_{\sigma_i, \sigma_j}}{n} \right)^{A_{ij}} \left( 1 - \theta_i \theta_j \frac{C_{\sigma_i, \sigma_j}}{n} \right)^{1-A_{ij}} = \prod_{i,j<i} \left( \theta_i \theta_j \frac{C_{\sigma_i, \sigma_j}}{n} \right)^{A_{ij}} + o\left( \frac{1}{n} \right)
$$

$$
= \prod_{(ij)\in\mathcal{E}} \theta_i \theta_j \frac{C_{\sigma_i, \sigma_j}}{n} + o\left( \frac{1}{n} \right).
$$

By making use of the Bayes theorem we can map the probability distribution of the labels to a physical analogue of spins interacting on the graph.

$$
\mathbb{P}(\boldsymbol{\sigma}|A) = \int d\boldsymbol{\theta} \, \mathbb{P}(\boldsymbol{\sigma}, \boldsymbol{\theta}|A) = \int d\boldsymbol{\theta} \, \mathbb{P}(A|\boldsymbol{\sigma}, \boldsymbol{\theta}) \frac{\mathbb{P}(\boldsymbol{\sigma})\mathbb{P}(\boldsymbol{\theta})}{\mathbb{P}(A)}
$$

$$
\underset{n\to\infty}{\sim} \frac{1}{\mathbb{P}(A)2^n} \prod_{(ij)\in\mathcal{E}} \frac{C_{\sigma_i, \sigma_j}}{n} \int d\boldsymbol{\theta} \, \mathbb{P}(\boldsymbol{\theta}) \theta_i \theta_j = \frac{1}{Z} \prod_{(ij)\in\mathcal{E}} C_{\sigma_i, \sigma_j} = \frac{1}{Z} e^{-\tilde{\mathcal{H}}(\boldsymbol{\sigma})}
$$

where we recovered the Boltzmann distribution with dimensionless Hamiltonian given by

$$
\tilde{\mathcal{H}}(\boldsymbol{\sigma}) = - \sum_{(ij)\in\mathcal{E}} \log \left[ C_{\sigma_i, \sigma_j} \right] \equiv - \sum_{(ij)\in\mathcal{E}} \mathrm{ath}\left( \frac{1}{r} \right) \sigma_i \sigma_j + const \tag{A.1}
$$

where $const$ is a constant that will be absorbed in the normalization factor. This last step gives rise to an Ising Hamiltonian. The following system of equations must then hold for some $r$:

$$
\log[c_{\mathrm{in}}] = \mathrm{ath}\left( \frac{1}{r} \right) + const. \tag{A.2a}
$$

$$
\log[c_{\mathrm{out}}] = -\mathrm{ath}\left( \frac{1}{r} \right) + const. \tag{A.2b}
$$

It is easy to check that $r = \zeta$ is the solution to this system of equations. From this result, one can then follow the derivation of the Bethe-Hessian matrix proposed in [2].

It has to be remarked that to obtain Equation (A.1) we neglected terms coming from non-nearest neighbours, in the limit for $n \to \infty$. The mapping is therefore not exact, but it still constitutes a useful tool to analyze and understand the problem.

Further note that, for disassortative networks, $c_{\mathrm{in}} < c_{\mathrm{out}}$ and thus $\zeta < 0$ as commented in Remark 3 in the main article. This would correspond to an anti-ferromagnetic interaction between the spins, in complete agreement with the mapping provided.

### B Mean and variance of the eigenvector

We need to identify the terms $\beta_i$ and $\mu_\alpha$ introduced in Assumption 1 to track the behavior of $\boldsymbol{\delta}$ and thus of the eigenvector $\boldsymbol{\sigma} + \boldsymbol{\delta}$. A first constraint on $\boldsymbol{\delta}$ follows from imposing the normalization of the eigenvector which, in the trivial limit equals $\boldsymbol{\sigma}$, the norm of which is $\sqrt{n}$. As such,

$$
\|(1 - \mu_\alpha)\boldsymbol{\sigma} + f_\alpha \boldsymbol{\beta} \odot \boldsymbol{N}\|^2 = n \tag{B.1}
$$

where $\boldsymbol{\beta} = (\beta_i)_{i=1}^n$, and $\boldsymbol{N}$ is a vector of zero mean and unit variance Gaussian random variables. Denoting $n\tilde{\beta}^2 \equiv \|\boldsymbol{\beta} \odot \boldsymbol{N}\|^2$ and observing that $\tilde{\beta} = O(\beta_i)$ – *i.e.* they have the same scaling with respect to $c$ –, we can rewrite this equation under the form:

$$
(1 - \mu_\alpha)^2 + f_\alpha^2 \tilde{\beta}^2 = 1. \tag{B.2}
$$

This provides a first relation between $\mu_\alpha$ and $\tilde{\beta}$. To obtain our next equations, we now explore boundary conditions on the model parameters in the limit of trivial clustering and at the phase transition where clustering becomes impossible.

It is established in [3] that there exists a critical value $\alpha_c \equiv 2/\sqrt{\Phi}$ for $\alpha$ below which community detection is (asymptotically) impossible. In particular, for $\alpha = \alpha_c$, the eigenvector $\boldsymbol{\sigma} + \boldsymbol{\delta}$ does not contain any information about the classes and thus $\mu_{\alpha_c} = 1$. From Equation (9), we then find that $f_{\alpha_c} = \sqrt{c\Phi - 1}/2$. Also, from (B.2), we get $\tilde{\beta} = 1/f_{\alpha_c}$. Updating (B.2), we now have an explicit expression for $\mu_\alpha$ for all $\alpha$. Recalling that $4f_\alpha^2 = \zeta_\alpha^2 - 1$ (from (6) and (9)) then gives

$$1 - \mu_\alpha = \sqrt{\frac{c\Phi - \zeta_\alpha^2}{c\Phi - 1}}. \tag{B.3}$$

Getting back to (7) and (8), it now remains to estimate $\beta_i$, which we shall perform in the limit $\alpha \to \sqrt{2c_{\text{in}}}$ of trivial clustering. To this end, combining both equations, we have

$$2f_\alpha(1 - \mu_\alpha)\sqrt{d_i}\tilde{N}_i - \zeta_\alpha \sum_{j \in \partial_i} f_\alpha \beta_j N_j + d_i f_\alpha \beta_i N_i = \lambda_\alpha[(1 - \mu_\alpha)\sigma_i + f_\alpha \beta_i N_i]$$

for $\tilde{N}_1, N_1, \ldots, \tilde{N}_n, N_n$ all (non necessarily independent) standard normal random variables. The second left-hand side term is proportional to $\sqrt{d_i}$ (and thus of order $O(\sqrt{c})$) as per the weak independence assumption of the $N_k$'s (Assumption 1). Dividing both sides by $f_\alpha \sqrt{d_i}$ to equate terms of order $O(1)$, the right-hand side now scales as $\lambda_\alpha/(f_\alpha \sqrt{d_i})$. As noted in Remark 1, in the trivial clustering limit where $\alpha \to \sqrt{2c_{\text{in}}}$, $\lambda_\alpha \to 0$, but it is not clear whether the right-hand side (after division by $f_\alpha \sqrt{d_i}$) vanishes; we now investigate this term in detail. One may at first observe that, if $c_{\text{out}} = \epsilon c_{\text{in}}$ for $\epsilon \ll 1$, since $c$ typically scales like $d_i$, we obtain that $f_\alpha \sqrt{d_i} = \sqrt{\epsilon c_{\text{in}}/2} + O(\epsilon)$. Hence, if $c_{\text{in}} \gtrsim \epsilon^{-1}$, the right-hand side vanishes. But imposing this growth condition is in fact not even necessary. If $\lambda_\alpha \propto f_\alpha^\eta$ for some $\eta > 1$, we directly obtain a vanishing right-hand side term; in Section 3 we argued that $\eta = 2$ (see Claim 1).
Denoting $\sum_{j \in \partial_i} \beta_j N_j \equiv \langle\beta\rangle N \sqrt{d_i}$ for some $\langle\beta\rangle > 0$, we may then rewrite

$$2(1 - \mu_\alpha)\tilde{N}_i - \zeta_\alpha \langle\beta\rangle N + \sqrt{d_i}\beta_i N_i \to 0 \tag{B.4}$$

in the limit $\alpha \to \sqrt{2c_{\text{in}}}$. Besides, $\mu_\alpha \to 0$ while $\zeta_\alpha \to 1$. We already argued that $\beta_i$ (and thus $\langle\beta\rangle$), which is of the order of $\tilde{\beta}$, scales as $1/f_{\alpha_c} = O(c^{-1/2})$. Thus, in the limit of large degrees, the second term in (B.4) is negligible and the third of order $O(1)$. Equating the large degree limiting variances of the resulting equation finally gives

$$\beta_i = \frac{2}{\sqrt{d_i}}.$$

We now have the mean and the variance of each vector component and we can estimate the expression of the overlap. Considering a node with $\sigma_i = 1$ without loss of generality, in the large $c$ limit, we have the approximate classification error for node $i$:

$$\mathbb{P}_{\text{err}}^i \simeq \frac{1}{\sqrt{2\pi[f_\alpha\beta_i]^2}} \int_{(1-\mu_\alpha)}^\infty e^{-x^2/(2[f_\alpha\beta_i]^2)}dx = \frac{1}{2}\left[1 - \text{erf}\left(\frac{1}{\sqrt{2}[f_\alpha\beta_i]}(1 - \mu_\alpha)\right)\right].$$

From this, the expression of the overlap follows.

## C  Extension to more than two classes

In order to generalize the argument carried on for two classes, first we look into the following quantity

$$\mathbb{P}(\ell_i|\ell_j, A_{ij} = 1) = \frac{\mathbb{P}(\ell_i, \ell_j|A_{ij} = 1)}{\mathbb{P}(\ell_j|A_{ij} = 1)} = \frac{\iint d\theta_i d\theta_j \mathbb{P}(\ell_i, \ell_j, \theta_i \theta_j|A_{ij} = 1)}{\mathbb{P}(\ell_j)}$$

$$= \frac{\iint d\theta_i d\theta_j \mathbb{P}(A_{ij} = 1|\theta_i, \theta_j, \ell_i, \ell_j)\mathbb{P}(\ell_i)\mathbb{P}(\ell_j)\mathbb{P}(\theta_i)\mathbb{P}(\theta_j)}{Z\pi_{\ell_j}}$$

$$= \frac{\pi_{\ell_i}C_{\ell_i,\ell_j}}{c} = \frac{(\Pi C)_{\ell_i,\ell_j}}{c} = \frac{(C\Pi)_{\ell_j,\ell_i}}{c}$$

By repeating the same argument on the average behavior of the adjacency matrix we obtain:

$$\langle(A\boldsymbol{u}^{(p)})_i\rangle = \sum_{j \in \partial(i)} \langle u_j^{(p)}\rangle = \sum_{j \in \partial(i)} \langle v_{\ell_j}^{(p)}\rangle = d_i \sum_{\ell_j} \mathbb{P}(\ell_j|\ell_i, A_{ij} = 1)v_{\ell_j}^{(p)}$$

$$= \frac{d_i}{c}\sum_{\ell_j}(C\Pi)_{\ell_i,\ell_j}v_{\ell_j}^{(p)} = \frac{d_i}{c}(C\Pi v^{(p)})_{\ell_i} = \frac{d_i}{c}\tau_p v_{\ell_i}^{(p)} = d_i\frac{\tau_p}{c}u_i^{(p)}$$

from which the result unfolds. In the simulations on synthetic networks, the off-diagonal terms of the matrix $C$ are drawn from a uniform distribution $\mathcal{U}(c_{\text{out}} - f, c_{\text{out}} + f)$, the element $C_{11}$ is fixed to $c_{in}$ and all the other diagonal terms are determined to ensure $C\Pi \mathbb{1}_k = c\mathbb{1}_k$. The randomness will make the eigenvalues of $C\Pi$ non degenerate and there will not be a unique transition. The line $c_{\text{in}} - c_{\text{out}} = k\sqrt{c}$ indicates the approximated position of the transition.

In Figure 1 we report the spectrum of $B$ in the case of four classes, that shows that the largest isolated real eigenvalues of the matrix $B$ are $\tau_p$ for $1 \le p \le k$, followed by $c/\tau_p$ for $2 \le p \le k$. This result can be obtained analytically from the linearization of the belief propagation equations (see [4]).

Figure 1: Spectrum of $B$. In green the isolated real eigenvalues outside the bulk corresponding to $\{\tau_p \Phi\}$, in red those inside the bulk, corresponding to $\{\zeta_p = c/\tau_p\}$; in blue all the others. We used 4 clusters of equal size, $n = 5000$, $c_{\text{in}} = 20$, $c_{\text{out}} = 5$, $f = 1.5$ and $\theta_i \sim \theta = \mathcal{U}(3, 13)^4$.

Figure 2(a) displays the overlap as a function of the hardness of the problem and of the number of classes comparing our algorithm with [2], evidencing a strong advantage in terms of performance for our algorithm. The red square underlines the fact that the two methods coincide *only* at the transition when $k = 2$ and the latter algorithm pays a lot in terms of performance for $k > 2$, even close to the transition. Figure 2(b) shows how $\hat{k} = |\{p, \ v_p(\sqrt{c\Phi}) < 0\}|$ is a good estimator of the number of classes. With $k_d = |\{p, \ \tau_p > \sqrt{c/\Phi}\}|$ we denote the number of theoretically detectable clusters and plot the quantity $2(\hat{k} - k_d)/(\hat{k} + k_d)$, showing small disagreement only close to the transition. The recovery being asymptotically exact, this can be interpreted as a finite size effect.

Figure 2: (a) Overlap (color scale) as a function of the number of classes ($k$) and hardness of the problem for the proposed algorithm (left) and $H_{\sqrt{c\Phi}}$ (right). Here, $n = 10\,000$, $c_{\text{in}} = 4 \to 40$, $c_{\text{out}} = 3$, $f = 2/k$, $\theta_i \sim [\mathcal{U}(3, 13)]^4$. Averaged over 10 samples.
(b) Recovery ($2(\hat{k} - k_d)/(\hat{k} + k_d)$) as a function of $k$ and the hardness of the problem for the same parameters as (a).