[Reviews · NeurIPS 2019]

Reviewer 1



Main comments: On the positive side, the results appear interesting, and the heuristic derivations seem highly plausible. Overall the proposed approach could be an appealing solution for community detection. On the negative side, the paper is not very easy to parse, leaving a lot to the reader to guess about notation and details of explanations. The derivations are quite sketchy, and the case for the proposed approach would be much stronger if i) additional arguments were provided to justify these derivations, and ii) additional numerical experiments were reported to support the claims (eg of gaussianity of eigenvector entries). Also, a comparison with not just (spectral methods based on the) initial version of the Bethe Hessian and adjacency matrix were reported, but also with alternative methods, eg regularized Laplacians. Further comments: -derivation of eq(5) is very quick, some justification of the iid nature of 0-1 variables summing to d_i in the DC-SBM would help. -l120, 'with the expectation still for...': do you mean you are taking a conditional expectation? If so then better use the mathematical symbol E( . |A,d_i) instead. -l180-181: is nu_i(r) the i-th smallest eigenvalue of H_r? I could not find the definition of this quantity earlier in the text. -l214-215, 'the vector u^(p)... is then associated with the p-th smallest eigenvector...': why is that? Is is explained in the supplementary material? If so, provide a pointer to where the explanation is given. -supplementary material A, evaluation of P(A|sigma, theta): you are missing an asymptotically constant multiplicative term due to non-edges. This does not matter in the end as it is absorbed in the normalization constant, but that should be corrected anyway. -supplementary material B: statement tilde(beta)=O(beta_i) does not make much sense. More generally the whole justification of approximate gaussianity of eigenvector entries in this section is too sketchy, and more arguments are needed to make a convincing case. -I did not understand how to read figure 2(b); footnote number ^4 in the caption of figure 2 does not point to anything apparently.

Reviewer 2



This paper looks at the community detection problem when the underlying network is a degree corrected stochastic block model in the sparse regime (i.e. constant average degree as the number of nodes tend to infinity). This model has been extensively studied and the authors are still able to make a contribution. On the rigorous side, for 2 communities, [16] proves that for \alpha <\alpha_c (notation of the paper see equation (2)), it is impossible to recover the communities better than random guessing and [17] shows that for \alpha>\alpha_c, a spectral method based on the non-backtracking matrix will achieve positive overlap. The authors in [8] suggest to use a symmetric matrix that they named the Bethe Hessian instead of the non-backtracking matrix and demonstrate empirically that a spectral algorithm based on this matrix works very-well. In order to use the Bethe Hessian, on needs to specify an additional hyperparameter called r compared to the non-backtracking matrix. A default choice was proposed in [8] and the paper under review suggests another generic choice for this parameter. With this new choice of the parameter, the spectral algorithm gives a better overlap for the degree corrected SBM. Some mathematical intuitions for this new value is given in section 2. Ways to estimate it are described in section 3. Extensions to several uneven-sized communities are provided in section 4 and experimental validation is done in section 5. -------------- Post feedback: I've read the feedback of the authors. A point I realized in the discussion with other reviewers: I doubt the overlap obtained by your spectral method will be 'optimal'. Usually spectral methods are used as initialization for a greedy method improving the overlap. The only optimal result with a rigorous proof that I know is from Mossel Neeman and Sly in Belief propagation, robust reconstruction and optimal recovery of block models where they show that running a variant of BP with a good initialization is optimal well above the threshold for SBM. Hence your remark 2 is probably not correct and it would be nice to have experiments with BP.

Reviewer 3



This paper studies clustering/community detection in the stochastic blockmodel with degree inhomogeneous vertex degrees. Major advances have been made in recent years on optimal algorithms for clustering in sparse stochastic block models, including a series of works analyzing belief propogation methods and developing spectral algorithms based on nonbacktracking random walks. This paper studies a family of spectral methods, collectively called the Bethe Hessian, for community detection in sparse graphs sampled from the *degree-corrected stochastic block model*, a variant of the stochastic block model which allows for a variety of degree distributions. High degree vertices typically affect the spectrum of the adjacency matrix by introducing spurious eigenvalues, and so other spectral methods are needed to correct for the degree inhomogeneaty. The main result of the paper is a detailed heuristic derivation (somewhat statistical physics inspired (as is common in this area)) of the optimal choice of hyperparameter for the Bethe Hessian spectral method. This derivation is validated with some experiments on real-world networks. I have mixed feelings about this paper. I do not think that it can be accepted on the basis of the theory it presents -- while theoretical motivations for hyperparameter settings are important, I think on its own this is too narrow for NeurIPS acceptance (and it is a demerit that the arguments are not quite rigorous). On the other hand, it does propose an algorithm which seems to beat the state of the art for community detection in some real data -- clearly this is an important benchmark. I think the paper would be substantially improved if the experimental aspect were more deeply explored -- it would be good to compare on real data to e.g. the nonbacktracking operator, and also to demonstrate empirically that the proposed paramter setting for the Bethe Hessian is optimal, rather than only exhibit experiments on one other possible parameter setting. UPDATE: I have read the author response. I am glad to hear that experiments with other values of $r$ will be added to the paper. Thanks for pointing out that there exists a value of $r$ such that clustering with the Bethe Hessian should act like clustering with B; even looking at ref [8] briefly it is not clear to me how formal this relationship is.

Reviewer 4



I think this is an interesting paper, establishing a choice of the parameter in the Bethe Hessian matrix that is threshold-optimal for the sparse degree-corrected block model. This is novel up to my knowledge. The random matrix theory (establishing the value of the overlap) behind the work is interesting and non-trivial. I see two slight drawbacks: ** The paper is not very clear about the comparison with performance of the non-backtracking matrix spectral method for the same problem. From my understanding that method has the same threshold. Does it have a worse overlap? I think so, but could not find a clear statement about it in the paper. ** Remark 2 and related appendix in SM seems non-sensical to me. The authors use the works Bayes-optimality and Nishimori condition in a way different from the literature to which they refer without defining them properly. Clearly a spectral method will not be able to reach a Bayes-optimal overlap, such as degree-corrected belief propagation is conjectured to reach. The derivation in the Appendix is more than heuristic. The authors end up with a mapping into an Ising model that would not even maintain the right sizes of the groups if exact sampling was implemented for it.

[Author Response · NeurIPS 2019]

A main concern shared by reviewers #1 and #3 is the lack of mathematical rigour. We address this in two points:

- The proposed parametrization of $H_r$ should not be seen as a simple tuning of a hyper-parameter. It is on the contrary the only consistent possible choice, according to three arguments: i) $\zeta$ is the eigenvalue resulting from the linearization of BP around its trivial fixed point (see Eq. (11) of [9]) and the corresponding eigenvector $g$, once processed as $g_i^{\text{in}} = \sum_{i \in \partial j} g_{ij}$ satisfies $H_\zeta g^{\text{in}} = 0$, that is precisely the eigenvector studied in the present article; ii) the mapping to the Ising Hamiltonian, from which $H_r$ was derived in the first place ([8]), is consistent only at $r = \zeta$ as explained in Section A of the supplementary material; iii) choosing $r = \zeta$ enables a resilience to degree heterogeneity in the DC-SBM, as developed in Section 2.2. These three arguments are independent and lead to the same parameter, creating a deep connection between $B$ and $H_\zeta$ other than the graph Laplacian to which $H_\zeta$ tends in easy problems.

- We agree that most of our derivations are heuristic. This has nevertheless been the case in the last few years in this research domain: a first heuristic derivations of the results via different techniques – mainly based on tools from statistical physics – is a first necessary step before the mathematical formalization. The study of $B$ in the sparse regime is a hot topic in research, and technically very challenging, see e.g. [10], [15], [16]. As suggested by reviewer #2, even fewer results are available on $H_r$, and this work, to our knowledge, represents a first attempt to characterize its eigenvectors. Finally, we also point out that our work already triggered new mathematical research (Coste, Zhu 2019: arXiv:1907.05603) in which some of our claims were formally proved.

We now address more specific concerns raised by each reviewer.

- Reviewer#1. Thank you for your detailed review. We hope that your concerns are fully addressed.

  Comparison with regularized Laplacian techniques of, e.g., (Qin 2013). Based on simulations, these methods have comparable performance to ours (ours never being worse). The regularized Laplacian, however, relies heavily on the normalization of the rows of the matrix containing the eigenvectors. Such normalization appears very powerful and in practice effective for many matrices. We believe that such step is not completely justified because i) the study of $\|L_\tau - \mathscr{L}_\tau\|$ is relevant when such quantity is small compared to the eigengap (Joseph, Yu 2014), that is, for simple classification problems. No guarantee is given for harder scenarios. ii) the detectability threshold is not mentioned in (Qin 2013) and it represents instead a fundamental aspect of our algorithm. We are currently working on a precise description of the connection between these two spectral techniques.

  In Eq. (5) the result is the expectation of the sum of Bernoulli random variables. This has been clarified in the new version. The independence comes from the tree like approximation: neighbours of a same node belong to conditionally independent branches. This is a standard technique used in BP (Mezard 2009), also formalized in e.g. (Salez 2011).

  In l120 we are taking a conditional expectation. The notation has been clarified in the new version. $\nu_p$ is the $p$-th smallest eigenvalue of $H_r$, as defined in l177. In l214-215, the vector $u^{(p)}$ corresponds to the $p$-th largest eigenvalue of $C\Pi$, denoted with $\tau_p$, hence to the $p$-th smallest $\zeta_p = c/\tau_p$. Decreasing the value of $r$, starting from $r = \sqrt{c\Phi}$, the informative eigenvalues go from negative to positive, hitting zero at $\zeta_p$. The $k$-th smallest will be the first, so it will correspond to $\zeta_k$ then the others follow, so the the $p$-th corresponds to $\zeta_p$. This has been clarified in the new version.

  The statement $\tilde{\beta} = O(\beta_i)$ means that the random variable $\beta_i$ has the same scaling (with respect to the average degree) than its expectation. The argument of Gaussianity is an assumption made on reasonable intuitions to conclude the calculus and is to be tested on the expression of the overlap compared to the simulations. Given the very good agreement, we understand that the approximations made to that point are reasonable and justify the description we made on the shape of the informative eigenvector.

  Figure 2 supp mat: $\hat{k}$ represents the number of classes estimated from our algorithm, while $k_d$ is the number of classes that are theoretically detectable. The color scale plots the quantity $2(\hat{k} - k_d)/(\hat{k} + k_d)$ as a function of the actual number classes ($k \geq k_d$) and the hardness of the problem. When this quantity is zero (white), the algorithm has detected all the detectable classes. In the caption $\mathcal{U}(\cdot)^4$ stands for uniform distribution raised to power 4.

- Reviewer #2. Thank you for your very positive review.

  We agree that the term $(r^2 - 1)I_n$ doesn't affect the spectral properties, but it is necessary to make the connections with $B$ and $H_r$. In order not to introduce a further matrix $(D - rA)$, we chose to write everything in terms of $H_r$ for the sake of clarity. Your interpretation about the kernel is correct: in the new version this has been pointed out.

  To estimate $\zeta_p$ we need to recompute each time the first $p$ eigenvalues (not the whole spectrum). This method is self contained in terms of $H_r$, but using the eigenvalues of $B'$ ([9]) can be an alternative way to estimate $\zeta_p$. We are currently working on an alternative and faster solution, based on a polynomial approximation.

- Reviewer #3. Thank you for your review. We hope that your concerns are fully addressed.

  The comparison of the performances for different values of $r$ is certainly interesting and has been added to the new version. Note however that the spectral algorithm on the matrix $B$ corresponds to $H_{(c_{\text{in}} - c_{\text{out}})\Phi/2}$ and has been seen in the literature (see e.g. [8]) to underperform the $H_{\sqrt{c\Phi}}$, consistently with the fact that $(c_{\text{in}} - c_{\text{out}})\Phi/2$ is farther away from $\zeta$ then $\sqrt{c\Phi}$.

[Meta-Review · NeurIPS 2019]

This submission was thoroughly reviewed and discussed among the reviewers. The conclusion is that the paper contains some interesting ideas and overall deserves acceptance but this was not an unanimous case as the theory part is thin and the justification of 'better performance' is moderate. Please further justify the latter, e.g. by comparing numerically the obtained overlap not only with the classical BH but also with the optimal overlap (see papers of Mossel-Neeman-Sly 13, Deshpande-Abbe-Monatanari 15 and Coja-Oghlan-Krzakala-Perkins-Zdeborova 16).